# LC-MS profiling and cytotoxic activity of *Angiopteris helferiana* against HepG2 cell line: Molecular insight to investigate anticancer agent

**Bipindra Pandey**[1], **Shankar Thapa**[1]*, **Mahalakshmi Suresha Biradar**[2], **Bhoopendra Singh**[3], **Jaya Bahadur Ghale**[4], **Pramod Kharel**[5], **Prabhat Kumar Jha**[6], **Ram Kishor Yadav**[6], **Sujan Dawadi**[7], **Poojashree V.**[8]

1 Department of Pharmacy, Madan Bhandari Academy of Health Sciences, Hetauda, Nepal, 2 Department of Pharmaceutical Chemistry, Al-Ameen College of Pharmacy, Bengaluru, India, 3 Institute of Pharmaceutical Research, GLA University, Chaumuhan, U.P., India, 4 Karnali College of Health Science, Purbanchal University, Nepal, 5 National Academy for Medical Science, Purbanchal University, Nepal, 6 School of Health and Allied Science, Pokhara University, Pokhara, Nepal, 7 Asian College for Advance Studies, Purbanchal University, Lalitpur, Nepal, 8 Department of Pharmacology, KLE College of Pharmacy KAHER, A Constituent Unit of KAHER-Belagavi University, Rajajinagar, Bengaluru, Karnataka, India

* tshankar551@gmail.com

**Data Availability Statement:** All the date used in manuscript are provided in within manuscript.

## Abstract

Liver cancer is one of the most prevalent malignant diseases in humans and the second leading cause of cancer-related mortality globally. *Angiopteris helferiana* was mentioned as a possible anticancer herb according to ethnomedicinal applications. However, the molecular docking and chemical profiling of the bioactive phytoconstituents accountable for the reported anticancer action still require research. The present study aims the phytochemical profiling and bioactivity evaluation of *A. helferiana*. The study design with in-vitro and in-silico technique of the LC-MS followed by a study of the ligand–protein interaction using the molecular docking method, and investigates the cytotoxic activity by MTT assay of *A. helferiana* bioactive compounds on HepG2 cell lines. LC-MS results detected seventeen phytoconstituents in *A. helferiana* extract belonging to variable chemical classes with most prevailing compounds such as Vicenin 1, Schafroside, Violanthin, Coumarin, Quercetin, Angiopterioside, and Corosolic acid. The finding concluded that Quercetin showed significant binding energy of -8.8 kcal/mol and then Schafroside also possesses the binding energy of -8.1 kcal/mol against the human PPAR-δ receptor (PDBID: 1I7G). The extract showed the moderate cytotoxic activity having $IC_{50}$ value of 236.93 μg/mL. Our finding suggests that these bioactive compounds could be developed as promising anticancer agent, but further in-vivo study require to validate the finding along with isolation of individual phytoconstituents.

**Funding:** The author(s) received no specific funding for this work.

## 1. Introduction

Liver is vital organ which is most susceptible to harm from exposure to xenobiotics (drugs, alcohol, illicit drugs, environmental toxins, and others), leading to a very high incidence of liver diseases [1]. Worldwide and in Nepal, the death rates from cancer, cirrhosis, fatty liver, and chronic hepatitis are high. Liver cancer is one of the most common malignant cancers in humans and the second largest cause of cancer-related death worldwide. It is a serious issue, particularly in less developed countries. An estimated 50% of medications were obtained through the use of herbal products. Ultimately, cancerous cells cause damage to the tissues and cause damaged to sudden cell divisions that multiply in healthy body cells, organ dysfunction and even death may be the result of these injured tissues [2, 3].

The main sources of new plant-based natural products with potential anticancer properties are their ethnomedical applications. Natural products are the main source of potent anticancer medications with innovative structures and distinct modes of action for the treatment of different types of cancer. Many plant-derived phytoconstituents have been used to treat hepatocellular carcinoma (HCC), including saponins, alkaloids, flavonoids, terpenes, and polysaccharides [4, 5]. Every segment of the global population is impacted by HCC. With the second highest death rate among all cancers, HCC is ranks as the fifth universal cancer among other cancers [6]. Annually, liver cancer is predicted to affect over a million people globally by 2025, making it a major global health concern. Ninety percent of cases of liver cancer are HCC, which has a high morbidity and mortality rate [7, 8].

Recent technological developments have made tandem analytical techniques such as; Liquid chromatography-mass spectroscopy (LC-MS), LC-MS/MS, Liquid chromatography-nuclear magnetic resonance (LC-NMR), and LC-NMR/MS are available [9, 10]. A potent method for characterizing target phytoconstituents in intricate plant extracts is tandem mass spectrometry (MS/MS). When compared to other dereplication techniques due to high sensitivity, selectivity, and quick screening capabilities via the online identification of secondary metabolites in plant extracts, the LC-MS/MS technique's is used [11]. Furthermore, the amount of time, money, and effort needed to screen natural products for biological activity has decreased significantly thanks to developments in computational biology [12, 13]. Nowadays, a lot of researcher use molecular docking to predict the binding affinity and mode of a drug-like molecule into the receptor's active site [14]. Virtual screening of a vast number of natural and synthetic compounds for activity against a variety of targets could save time, effort, and provide a quick expectation for the most promising anticancer candidates [15].

*Angiopteris helferiana* (*A. helferiana*) *C. Presl* is a large fleshy fern that is a member of the Marattiaceae family and can be found in moist forests in Southeast Asia, China, Nepal, India, and Sri Lanka at elevations between 900 to 1400 meters. In Nepal, the rhizome of *A. helferiana* is traditionally used to treat fatigue, muscle and bone pain [16]. The rhizome of this plant is also reportedly used in the Konkan region of Maharashtra, India, to treat scabies. In Bangladesh, the rhizome of *A. helferiana* was used in dysentery, infections, scabies, and muscular aches. Similarly, *A. helferiana* leaf is used ethnomedicinally to treat bloating and hair loss by making its topical powder and oral soup [17].

Prior research on the light yellow-colored rhizome of *A. helfiarana* revealed biological activity including hepatoprotective activity, anti-inflammatory, anti-obesity, and antidiabetic properties [18]. Additionally, Angiopteroside and (−)-epi-Osmundalactone, two lactones, were also isolated from the same cultivars [16]. The different species of *Angiopteris* show significant anticancer activity [19]. More precisely, S. Sara and R. Ruby explored the antiproliferative effect of *Angiopteris evecta* against cultured HT-29 colon cancer cells in their book chapter [20]. Similarly, S. Nur et al. [21], reported the significant cytotoxic effect of *Angiopteris ferox*

against different three cancer cell line. A. Aisyah et al., reported cytotoxic effect of methanolic fraction of *Angiopteris ferox* against pulmonary HTB cell line with an $IC_{50}$ grade of 78.96 μg/mL [22]. These data make us curious to explore the cytotoxic activity of *A. helferiana* against hepatocellular carcinoma (HepG2 cell line). Moreover, to date, there have been no reports on the cytotoxic activity, LC-MS/MS profiling, and in-silico studies from the local cultivars (Rato Gai-khurey) of *A. helferiana*. In this study, Liquid chromatography-mass spectroscopy (LC-MS) profiling, Fourier transform infrared spectroscopy (FTIR) analysis, cytotoxic activities and computational studies of bioactive compounds from *A. helferiana* were carried out, which might lead to the potential management of hepatocellular carcinoma.

## 2. Results and discussion

### 2.1. FT-IR spectra interpretation

The functional groups of the extract of *A. Helferiana* were determined using FT-IR spectroscopy. The spectrum shows the characteristic absorption peak at the range of 3500 to 3250 cm$^{-1}$. The peak was broad having medium intensity which indicate the presence of hydrogen bonded alcoholic or phenolic group in the extract (Fig 1). Moreover, the presence of sharp peak at 2927cm$^{-1}$ confirms the presence of aliphatic C-H group. A medium sharp absorption peak was observed at 2362 cm$^{-1}$ indicates the presence of C≡N group. It is noteworthy to mention that, the presence of carbonyl group (C = O, stretching) is supported by the strong absorption peak in the range of 1703 to 1627 cm$^{-1}$. Furthermore, a strong-intensity absorption peak at 1054 cm$^{-1}$ confirms the presence of ester (C-O, stretching) functional group. Other noticeable peaks are presents around 1392 cm$^{-1}$ and 1269 cm$^{-1}$. A peak at 1392 cm$^{1}$ was likely due to the C-H bending vibrations from methyl groups or potentially symmetric stretching from a nitro group. Similarly, a peak at 1269 cm$^{1}$ was likely due to the C-O stretching vibrations from an ether group or deformation from a nitro group. The bending and stretching vibration from

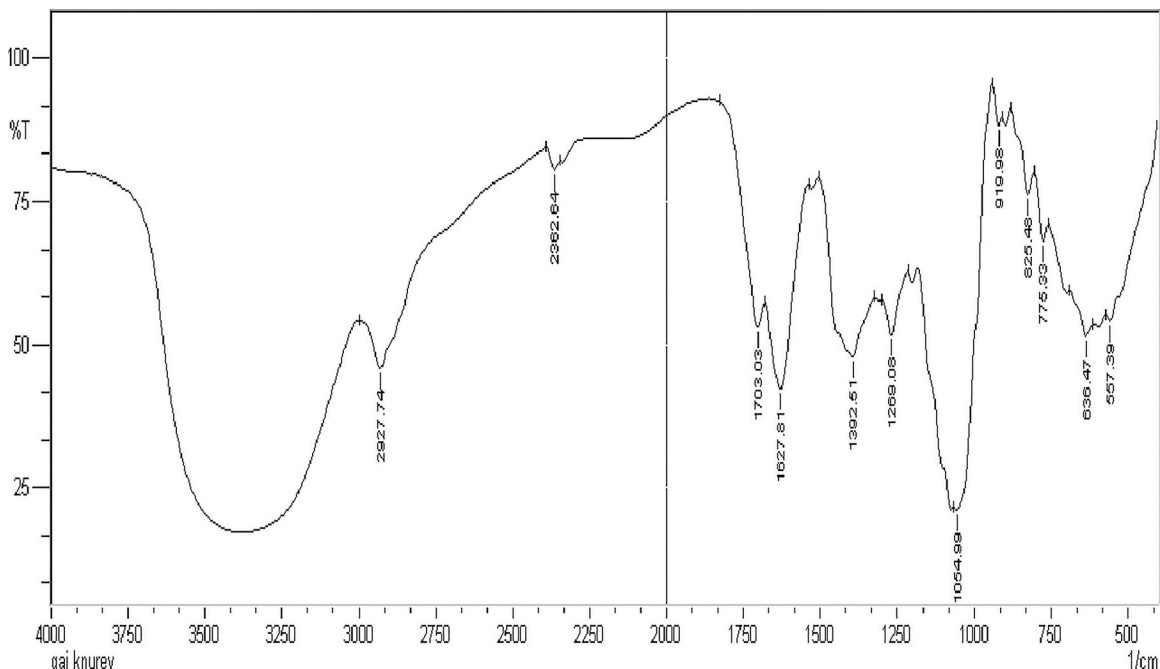

**Fig 1. FT-IR spectra of *Angiopteris helferiana*.**

900 cm$^{-1}$ to 500 cm$^{-1}$ represents the finger printing region of *A. helferiana* extract. Finger print region is the distinct characteristic region for the sample which is unique for each compound [23].

## 2.2. LC-MS profiling

Untargeted LC-MS analysis was conducted to examine and provisionally identify the isolated metabolites. The LC-MS analysis of an ethanolic extract detected several substances. The secondary metabolites were identified through the use of LC retention time, molecular mass and high-resolution mass spectra analysis in the extracts (Fig 2). We detected the peaks in LC-MS chromatograms by utilizing mass spectra m/z databases including Mass Bank of Europe (https://massbank.eu/MassBank/) and National Library of Medicine (https://www.nlm.nih. gov/), as well as relevant literature [24–27]. The discovered peaks are labelled as tentative due to the presence of natural products in isomeric forms, including isomerization of aglycones (e.g. flavones and isoflavones), or as isobaric compounds with the same molecular weight but distinct elemental composition [28].

A total of 17 phytoconstituents were found in the extracts, consisting of phenolic acids, flavonoids, and polyphenols (Figs 2S to 15S in S1 File). We also detected minor number of amino acids, ascorbic acid, and trace element but excluded from the analysis. The precursor ions at m/z 564 and 578 identified the Flavonoid glycoside clusters as Vicenin 1, Isoschaftoside, Schafroside, and Violanthin, respectively. The flavonoids were identified at m/z 146 and 302, corresponding to clusters of Coumarin and Quercetin, respectively. Additionally, from precursor ions at m/z 290 and 472 suggests natural product, indicating the presence of Angiopteroside and Corosolic acid (Table 1).

## 2.3. Protein validation

The PROCHECK service facilitates in assessing the three-dimensional geometry of protein structures by categorizing residues into different colours based on their areas: red (preferred), yellow (additionally allowed), pale yellow (generously allowed), and white (disallowed) [29], as illustrated in Fig 3. Ramachandran plots for all residue types are displayed in Fig 16S in S1 File. Human PPAR-δ protein contains 259 residues, with 217 (92.3%) of them located in the most liked area (A, B, L). Approximately 18 residues (7.7%) were found in the additional authorized regions (a, b, l, p). None of the residues located in generously allowed regions and residue in

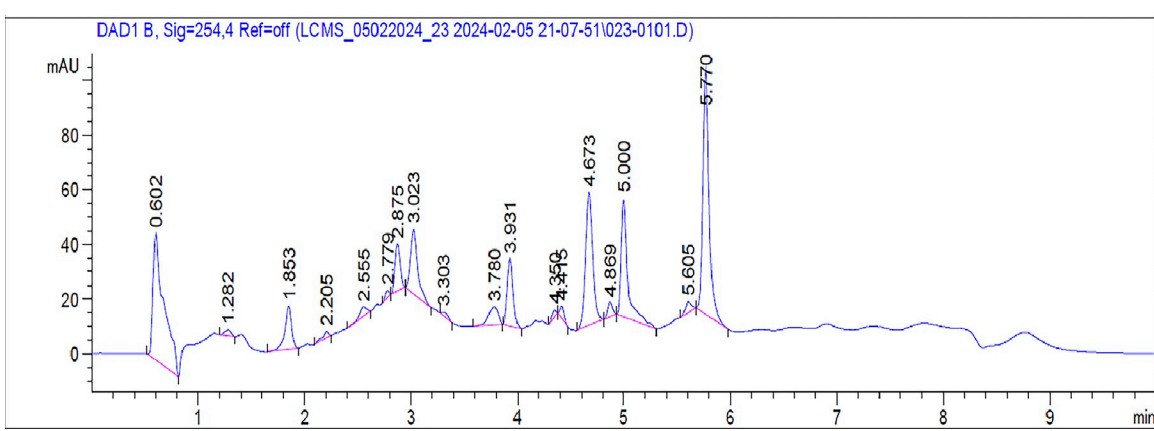

**Fig 2. LC-MS spectra of *Angiopteris helferiana* extract.**

**Table 1. LC-MS data-positive ion mode for tentative compounds identification in plant extract.**

| SN | RT | Formula | MS ion (m/z) [M+H] | MS/MS (m/z) fragment ions | Tentative Identification | Area (%) | Chemical classification |
|---|---|---|---|---|---|---|---|
| 1 | 0.602 | $C_7H_5FN_2$ | 136 | 122, 117, 100, 57 | 4-Fluro-1H-indazole | 20.531 | Heterocyclic compound |
| 2. | 1.853 | $C_7H_6O_4$ | 180 | 128, 113, 89, 57 | Acetyl salicylic acid | 4.396 | Organic aromatic compound |
| 3. | 2.20 | $C_{26}H_{28}O_{14}$ | 564 | 437, 324, 301, 205, 150, 149, 57 | Vicenin 1, Isoschaftoside, Schafroside | 0.560 | Flavonoid glycoside |
| 4. | 2.55 | $C_{27}H_{30}O_{12}$ | 578 | 463, 419, 375, 149, 143, 122, 57 | Violanthin | 1.130 | Flavonoid |
| 5. | 2.77 | $C_{20}H_{12}O_5F_2$ | 412 | 301, 283,282, 265, 177, 149, 97 | Oregon green | 0.458 | Organic aromatic compound |
| 6. | 2.87 | $C_{13}H_{18}O_7$ | 290 | 238, 211, 122, 117, 57 | Angiopteroside, Osmundalin | 3.786 | Glycoside |
| 7. | 3.02 | $C_{20}H_{22}O_2N_2$ | 324 | 207, 290, 217, 195, 128, 113, 89 | Quinine | 7.178 | Alkaloid |
| 8. | 3.30 | $C_{27}H_{30}O_{15}$ | 564 | 477, 383, 307, 237, 216, 118, 117, 115 | Vicenin 2, Vicenin 3 | 0.374 | Flavonoid glycoside |
| 9. | 4.35 | $C_{29}H_{45}O_4$ | 472 | 404, 347, 299, 238, 211, 122, 117, 57 | Corosolic acid | 0.422 | Pentacyclic triterpene |
| 10. | 4.67 | $C_9H_6O_2$ | 146 | 149, 122,118, 101, 100, 57, 90 | Coumarin | 14.823 | Coumarin |
| 11. | 4.86 | $C_{15}H_{12}O_6$ | 302 | 242, 175, 113, 69 | Quercetin | 1.038 | Flavonoid |
| 12. | 5.00 | $C_7H_{10}O_2$ | 100 | 89, 70, 57 | Osmundlactone | 11.546 | Heterocyclic compound |
| 13. | 5.60 | $C_{27}H_{30}O_{14}$ | 578 | 418, 395, 339, 311, 249, 133, 113, 69 | Isoviolanthin | 0.908 | Flavonoid glycoside |

Note: RT = Retention time; SN = Serial number.

banned regions (Fig 3). Within the human PPAR-δ protein, 235 residues were non-proline and non-glycine. Proline and glycine residues are depicted as triangles, with 9 and 10 occurrences, respectively. Fig 16SB in S1 File displays the hydrophobicity map of the human PPAR-δ protein in the study. The amino acid residues are indicated in brackets, with those in poor conformations (score < -3.0) labelled. Shaded regions represent favorable conformations observed in163 structures analyzed at a resolution of 2.0Å or higher. The shading represents advantageous conformations identified from an examination of 163 structures with a resolution of 2.0Å or higher.

To be considered stable, a protein's instability index must be less than 40 [30]. It was determined to be 34.08 in our instance, while the theoretical isoelectric constant was 5.86. It was discovered that the protein's grand average hydropathy value (GRAVY) was -0.170. There are more details in the Table 2.

## 2.4. Molecular docking

Molecular docking study was conducted to predict the binding affinity and binding orientation of phytoconstituents of *A. helferiana* extract against human PPAR-δ receptor (PDBID: 1I7G). Molecular docking is a computational technique used in drug discovery and molecular biology to predict the preferred orientation of molecules [31] (phytoconstituents of *A. helferiana* extract) when bound to a target molecule (human PPAR-δ receptor). It involves simulating the interaction between the two molecules to predict the most energetically favorable binding pose [32]. Molecular docking is highly significant as it aids understanding the binding mechanisms between drugs and their target proteins, facilitating the design of novel therapeutic compounds [33]. By predicting the binding affinity and mode of interaction, molecular docking helps prioritize potential drug candidates, saving time and resources in the drug development process and contributing to the identification of new agent for various diseases [34].

The role of PPAR-δ in cancer is intricate and context-dependent. PPAR-δ, a nuclear receptor, plays a pivotal role in regulating various physiological processes, including lipid metabolism, inflammation, and cell proliferation [35]. Its involvement in cancer is characterized by

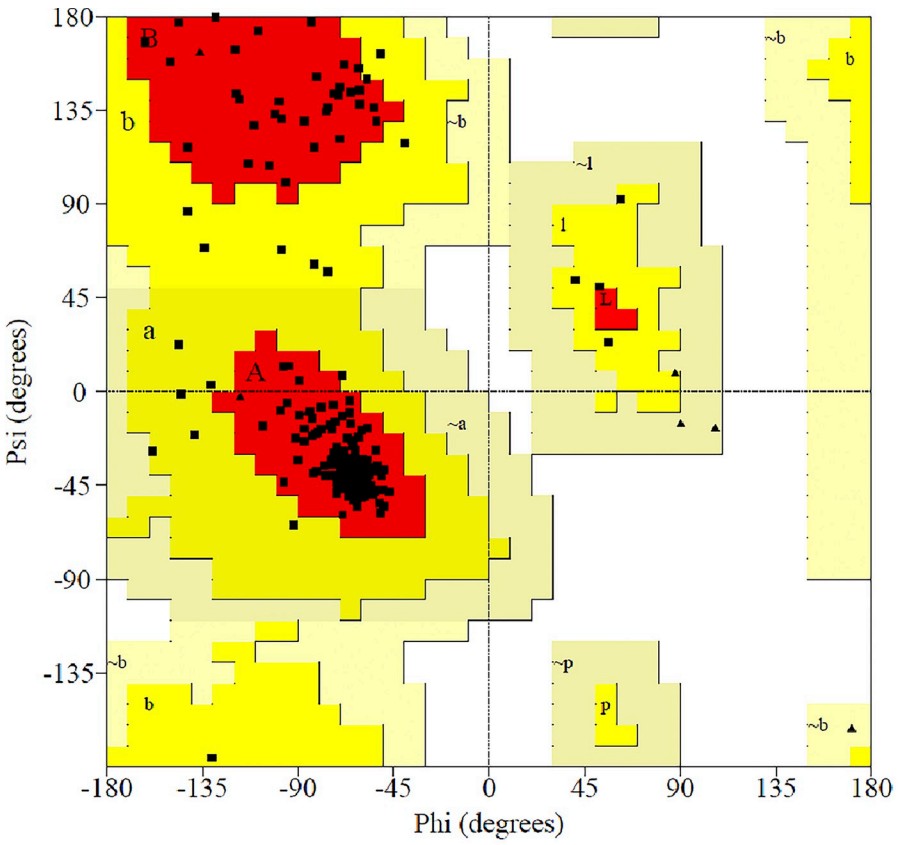

## Plot statistics

| | | |
|---|---|---|
| Residues in most favoured regions [A,B,L] | 217 | 92.3% |
| Residues in additional allowed regions [a,b,l,p] | 18 | 7.7% |
| Residues in generously allowed regions [~a,~b,~l,~p] | 0 | 0.0% |
| Residues in disallowed regions | 0 | 0.0% |
| | ---- | ------ |
| Number of non-glycine and non-proline residues | 235 | 100.0% |
| Number of end-residues (excl. Gly and Pro) | 5 | |
| Number of glycine residues (shown as triangles) | 10 | |
| Number of proline residues | 9 | |
| | ---- | |
| Total number of residues | 259 | |

**Fig 3. Ramachandran plot to validate the protein.**

both anti-tumorigenic and pro-tumorigenic effects, with outcomes varying based on tissue type and specific cellular contexts. Activating PPAR-δ has been associated with inhibiting cell proliferation and inducing apoptosis in certain instances, suggesting potential anti-tumorigenic effects [36, 37]. Additionally, its anti-inflammatory properties may indirectly hinder tumorigenesis by reducing chronic inflammation. On the other hand, PPAR-δ activation has

**Table 2. Physiochemical properties of human PPAR-δ protein.**

| Physiochemical properties | Human PPAR-δ (PDB ID: 1I7G) |
|---|---|
| Molecular Weight | 52225.13 |
| Molecular Formula | $C_{2303}H_{3655}N_{619}O_{695}S_{34}$ |
| Total no. of atoms | 7306 |
| Theoretical PI | 5.86 |
| Aliphatic index | 86.69 |
| Instability index | 54.08 |
| GRAVY | -0.170 |
| +R (Arg + Lys) | 52 |
| -R (Asp + Glu) | 61 |
| Ext. coefficient /Abs 0.1% (= 1 g/l) | 0.399 |

been linked to pro-tumorigenic effects, such as promoting angiogenesis and influencing lipid metabolism [38].

Around 17 phytoconstituents of *A. helferiana* extract were identified by LC-MS screening and subjected for docking. Site specific docking was utilized as it gives space to the ligands molecules to bind at validated specific site [39]. The docking results were sorted for phytoconstituents with least binding energy, maximum hydrogen bonding and minimum bond length. The interactions were visualised in 2D and 3D format included nature of bonding, bond length (Å) and respective binding energy in kcal/mol. Most prominent common amino acid participate for the interaction were Cys275, Cys277, Val332, Thr279, Tyr334, Met220 and Ala333. The prominent molecular interaction includes; conventional hydrogen bond, pi-pi stacked, pi-alkyl, carbon-hydrogen bond, pi-loan pair and pi-pi T shaped bond. Moreover, conventional hydrogen bonding is prominent with most of the interaction which indicate the possibility of stable ligand-protein complex. All the phytoconstituents showed moderate to good binding affinity ranges from -5.2 to -8.6 kcal/mol (Table 3). We have compared the test results with the native co-crystal ligand (**AZ242**) which showed good binding affinity of -7.7 kcal/mol. From phytoconstituents side, Quercetin (**B12**) and Schaftoside (**B7**) showed best binding affinity of -8.8 and -8.1 kcal/mol, respectively (Table 3). These two phytoconstituents exhibited better binding affinity than the native co-crystal ligand. However, Osmundlactctone (**B8**) showed the least binding affinity of -5.1 kcal/mol (Table 3).

Quercetin (**B12**) and Schaftoside (**B7**) both formed three conventional hydrogen bonds with Thr283, Glu286, Ser323 and Cys275, Thr279, Glu282, respectively (Figs 4 and 5). Quercetin (**B12**) had a lower binding energy compared to the co-crystal, indicating that it has a stronger affinity for the target than the co-crystal native ligand (Fig 6). Schaftoside (**B7**) had a shortened bond length of less than 2.5 Å, particularly for conventional hydrogen bonds, which enhances stability. Quercetin (**B12**) had a lower binding energy (better binding affinity) but showed a greater bond length between interacting amino acids and functional groups (more than 2.5 Å) compared to Schaftoside (**B7**). Increased bond distances could result in an unstable protein-ligand combination. Although the binding affinity of Quercetin (**B12**) was reported excellence but its stability issue, Schaftoside (**B7**) was consider best phytoconstituent to recommend for further investigation. Therefore, we recommend Schaftoside (**B7**) as a suitable candidate for more research. The 2D molecular interaction of other phytoconstituents is given in Figs 17S to 27S in S1 File.

## 2.5. ADME properties

The ADMETlab 2.0 server was utilized to investigate the ADME characteristics of 17 phytoconstituents. The phytoconstituents exhibited in the range of high to low gastrointestinal

**Table 3. Molecular insight, binding site amino acid, binding affinity of phytoconstituents of *A. helferiana* against human PPAR-δ receptor (PDBID: 1I7G).**

| SN | Compound name (code) | Binding site amino acid | Nature of interaction | Binding affinity (ΔG) | RMSD value (in Å) |
|---|---|---|---|---|---|
| 1. | Violanthin (**B1**) | Arg226, Asn326, Lys222, Glu369 | Conventional hydrogen bond, Pi-Pi stacked, Pi-alkyl, Carbon-hydrogen bond, Pi-loan pair and Pi-pi T shaped bond | -7.1 | 2.451 |
| 2. | Vicenin 1 (**B2**) | Cys (275, 276, Leu245, Glu282, Thr279, ala333, Val332 | | -7.8 | 2.988 |
| 3. | Vicenin 2 (**B3**) | Cys (275, 276), Val332, Thr279, Glu282 | | -7.6 | 1.826 |
| 4. | Vicenin 3 (**B4**) | Cys (275, 276), Val332, Thr279, Glu282, Ala333 | | -7.7 | 2.806 |
| 5. | Isoschaftoside (**B5**) | Cys275, Val332, Leu254, Thr279, Ala333 | | -7.8 | 2.730 |
| 6. | Isoviolanthin (**B6**) | Pro357, Asp353, Glu439, Asp360 | | -7.1 | 2.874 |
| 7. | Schaftoside (**B7**) | Cys275, Val324, Glu282, Tyr334, Met220, Val332, Thr279, Met220 | | -8.1 | 2.727 |
| 8. | Osmundlactctone (**B8**) | Ile354, Cys276, Phe273, Tyr (314, 464), His440 | | -5.2 | 1.979 |
| 9. | Angiopteroside (**B9**) | Asn219, Met (220, 320), Phe218, Glu286, Tyr334 | | -7.6 | 2.627 |
| 10 | Corosolic acid (**B10**) | Arg388, Pro389, Tyr311, Thr438, Val437 | | -7.3 | 2.809 |
| 11. | Coumarin (**B11**) | Ser280, Gln277, Cys276, Val444, His440 | | -6.6 | 1.474 |
| 12. | Quercetin (**B12**) | Thr283, Glu286, Ser323, Tyr334, Met320 | | -8.8 | 2.165 |
| 13. | Quinine (**B13**) | Tyr (214, 334), Met (220, 320, 330), Phe218, Leu321 | | -7.6 | 2.511 |
| 14. | Acetyl salicylic acid (**B14**) | Met220, Val324 | | -6.0 | 1.740 |
| 15. | 4-fluro-1H-indazole (**B15**) | Ser280, Tyr314 | | -6.9 | 0.987 |
| 16. | Oregon green (**B16**) | Pro389, Arg434, Met467, Glu315, Val437, Tyr311, Arg388 | | -7.3 | 2.582 |
| 17. | Osmundalin (**B17**) | Glu286, Asn221, Asp372, Ser323 | | -7.4 | 2.565 |
| 18. | Co-crystal ligand (**AZ 242**) | Cys275, Thr279, Ala334, Ile317, Leu321, Val332 | | -7.7 | |

absorption (GIA), with probability score of 0.1–1 (symbol—and +++) [40]. Phytoconstituents **B8** and **B10**-**B16** belongs to the category 0 which indicates the human intestinal absorption (HIA) greater or equal to 30%. Remaining 9 phytoconstituents (**B1**-**B7**, **B9** and **B17**) showed the HIA value less than 30% belonging to the category 1 (Table 4). The concentration of a substance in the nonpolar phase divided by its concentration in the aqueous phase at equilibrium is known as the partition coefficient (P). The LogP value is obtained by taking the logarithm of this ratio. LogP, or the logarithm of the partition coefficient, is a metric utilized in drug development to evaluate the lipophilicity or hydrophobicity of a substance. The compound's partitioning tendency between an aqueous phase (often octanol) and a nonpolar phase is quantified [41]. Log of the octanal/water partition coefficient indicates that most of phytoconstituents don't align with the optimal reference value of 0 to 3. However, the compound having high binding affinity (**B12**) showed the optimal LogP value of 2.15. Only Corosolic acid (**B10**) was insoluble (-4.53) in water. Remaining phytoconstituents were sparingly (-2 to -4) to freely soluble (0 to -2) in water. All these three parameters determine the absorption of phytoconstituents through gastrointestinal tract. All the phytoconstituents can distribute easily into the body

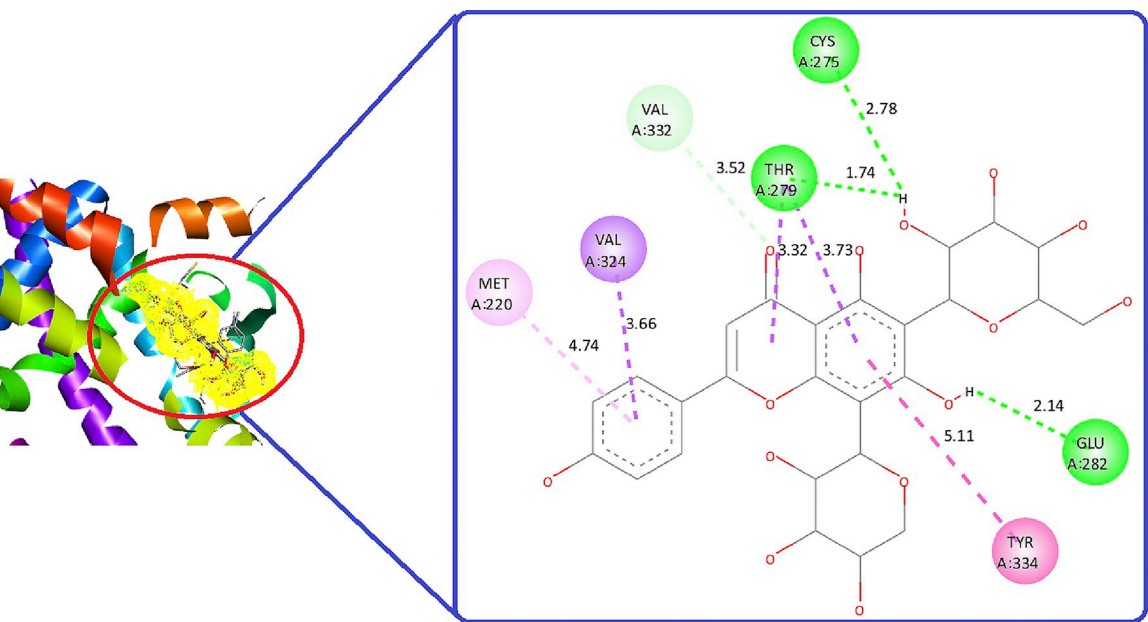

**Fig 4. 2D and 3D molecular interaction of Schaftoside (B7) against PPAR-δ (PDBID: 1I7G).**

compartment as all the phytoconstituents showed the optimal volume of distribution (VD) value (0.04–20 L/kg). The volume of VD determines the half-life and dosing frequency of the drug. If a drug has high VD, more of the drug is distributed in tissue than plasma. Metabolism of drugs is a complex biotransformation process, where drug molecules are get converted into their metabolites by various metabolizing enzymes [42]. The data provided in the table

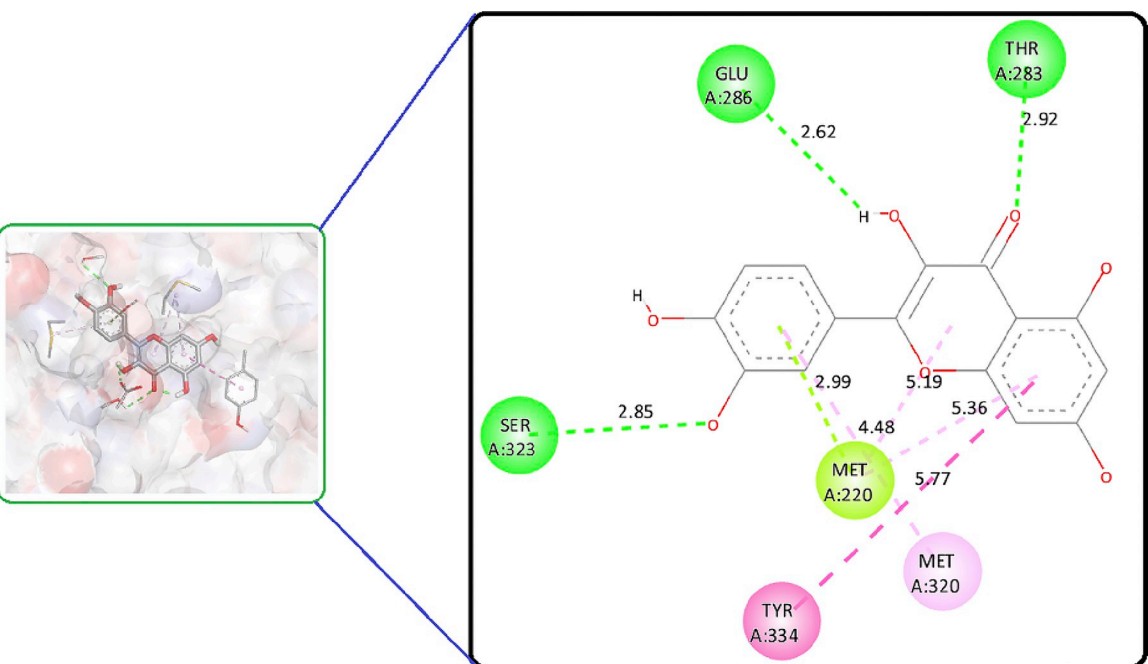

**Fig 5. 2D and 3D molecular interaction of Quercetin (B12) against PPAR-δ (PDBID: 1I7G).**

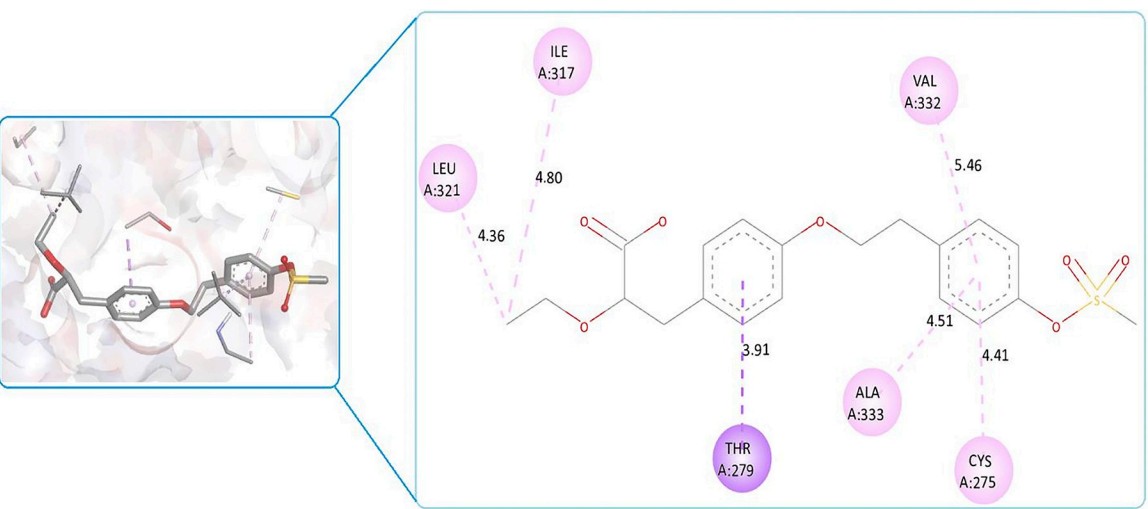

**Fig 6. 2D and 3D interaction of co-crystal native ligand (Az 242) against PPAR-δ (PDBID: 1I7G).**

predicts whether the compound is likely to be metabolized by $P_{450}$ or not. Most of the phytoconstituents did not inhibit the CYPA2 enzyme except **B11**, **B12**, and **B15**. Phytoconstituents **B8**, **B11**, **B12** and **B15** had moderate elimination rate (5–15 mL/min./kg) whereas remaining phytoconstituents has low renal clearance (less than 5 mL/min./kg). The predicted half-life ($T_{1/2}$) of all the phytoconstituents was low (less than 3 hours) (Table 4).

## 2.6. Drug likeliness properties

The web server ADMETlab 2.0 was utilized to assess drug likeliness attributes based on Lipinski's Parameter. The drug likeliness was assessed based on 2D molecular descriptors such as

**Table 4. ADME properties of phytoconstituents.**

| Com. code | Absorption | | | Distribution | Metabolism | Elimination | |
|---|---|---|---|---|---|---|---|
| | LogP | LogS | HIA | VD | CYPA2 inh. | CL | $T_{1/2}$ |
| B1 | 0.28 | -2.29 | +++ | 0.72 | — | 1.35 | 0.475 |
| B2 | -0.05 | -2.39 | +++ | 0.74 | — | 1.23 | 0.55 |
| B3 | -0.33 | -1.60 | +++ | 0.69 | — | 1.25 | 0.64 |
| B4 | -0.33 | -1.60 | +++ | 0.69 | — | 1.25 | 0.64 |
| B5 | -0.06 | -2.62 | +++ | 0.75 | — | 0.75 | 1.26 |
| B6 | 0.37 | -2.17 | +++ | 0.75 | — | 1.39 | 0.51 |
| B7 | -0.24 | -2.10 | +++ | 0.79 | — | 1.28 | 0.55 |
| B8 | 0.20 | 0.50 | — | 0.85 | — | 5.50 | 0.84 |
| B9 | -1.05 | -0.37 | ++ | 0.51 | — | 1.73 | 0.84 |
| B10 | 5.54 | -4.53 | — | 0.59 | — | 2.12 | 0.12 |
| B11 | 1.67 | -2.01 | — | 0.83 | +++ | 10.58 | 0.72 |
| B12 | 2.15 | -3.67 | — | 0.57 | +++ | 8.28 | 0.92 |
| B13 | 2.21 | -2.63 | — | 2.19 | - | 1.81 | 0.57 |
| B14 | 1.23 | -1.65 | — | 0.23 | — | 2.73 | 0.91 |
| B15 | 2.01 | -2.43 | — | 1.64 | +++ | 9.95 | 0.38 |
| B16 | 4.00 | -3.77 | — | 0.70 | — | 0.91 | 0.57 |
| B17 | -1.71 | -0.39 | ++ | 0.47 | — | 1.63 | 0.64 |

**Table 5. Drug likeliness properties of phytoconstituents.**

| Com. code | MW | CaCO₂ per. | TPSA | HBA | HBD | nRB | LR 5 |
|:---:|:---:|:---:|:---:|:---:|:---:|:---:|:---:|
| B1 | 578.16 | -6.40 | 250.97 | 14 | 10 | 4 | Violation |
| B2 | 564.15 | -6.38 | 250.97 | 14 | 10 | 4 | Violation |
| B3 | 594.16 | -6.42 | 271.20 | 15 | 11 | 5 | Violation |
| B4 | 594.16 | -6.42 | 271.20 | 15 | 11 | 5 | Violation |
| B5 | 564.15 | -6.37 | 250.97 | 14 | 10 | 4 | Violation |
| B6 | 578.16 | -6.39 | 250.97 | 14 | 10 | 4 | Violation |
| B7 | 564.15 | -6.37 | 250.97 | 14 | 10 | 4 | Violation |
| B8 | 100.00 | -4.68 | 46.53 | 3 | 1 | 0 | Obey |
| B9 | 290.10 | -5.69 | 125.68 | 8 | 4 | 3 | Obey |
| B10 | 472.36 | -5.44 | 77.76 | 4 | 3 | 1 | Obey |
| B11 | 146.04 | -4.58 | 30.21 | 2 | 0 | 0 | Violation |
| B12 | 302.04 | -5.20 | 131.36 | 7 | 5 | 1 | Obey |
| B13 | 324.18 | -4.97 | 45.59 | 4 | 1 | 4 | Obey |
| B14 | 180.04 | -5.06 | 63.60 | 4 | 1 | 3 | Obey |
| B15 | 136.04 | -4.38 | 28.68 | 2 | 1 | 0 | Obey |
| B16 | 412.04 | -4.94 | 125.04 | 7 | 3 | 3 | Obey |
| B17 | 290.10 | -5.74 | 125.68 | 8 | 4 | 3 | Obey |

molecular weight (MW), hydrogen bond donor (HBD) count, hydrogen bond acceptor (HBA) count, rotatable bond (RB), Log P value, and molecular surface area (MSA). Lipinski's Rule states that "for a compound to have good oral bioavailability, it should have a molecular weight of less than 500 Da, fewer than 10 hydrogen bond acceptors, fewer than 5 hydrogen bond donors, and an octanol/water partition coefficient of less than 5" [43]. Topological polar surface area (TPSA) is an acronym for topological polar surface area, a molecular descriptor utilized in drug design to evaluate the molecular size and complexity of a substance. It offers statistics on the molecule's accessible surface area based on the quantity and variety of atoms it contains. The TPSA value serves as a parameter in drug-likeness assessments. Distinguishing between drug-like compounds and larger, more complex molecules with higher molecular weight and structural complexity can provide hurdles in development and optimization [44].

CaCO₂ permeability plays an important role for drug absorption. The phytoconstituents B**8**, **B11**, **B13**, **B15** and **B16** showed the optimal CaCO₂ permeability value (higher than −5.15 cm/s), but remaining 12 phytoconstituents showed a value lower than the optimal range (Table 5). Along with other phytoconstituents (**B1**-**B7** and **B11**), Schaftoside (**B7**) violate the Lipinski's rule of five whereas, Quercetin (**B12**) obey the rule. Furthermore, phytoconstituents **B8**, **B9**, **B10**, **B13**, **B14**, **B15**, **B16**, and **B17** also obeyed the rule (Table 5). All the parameters are in favour towards the Quercetin (**B12**) to choose as best anti-cancer agent. But the molecular docking report suggest that, the unstable nature of Quercetin (**B12**) binding against receptor hinder the therapeutic efficacy of drug. Therefore, we consider Schaftoside (**B7**) could be the potential candidate as anti-cancer agent. Schaftoside (**B7**), part of phytoconstituent of plant *A. helferiana*, has demonstrated promising drug potential despite violating Lipinski's Rule of 5. With a molecular weight of 564.15 Da, a CaCO₂ permeability score of -6.37, and a TPSA of 250.97, Schaftoside (**B7**) exceeds the preferred limits for several key drug-likeliness parameters. Additionally, it possesses 14 hydrogen bond acceptors, 10 hydrogen bond donors, and 4 rotatable bonds, all contributing to its multiple Lipinski's rule violations. However, the standout feature of Schaftoside (**B7**) is its impressive molecular docking score of -8.1 kcal/mol, indicative of a strong binding affinity to its target protein. Most importantly the Schaftoside

(**B7**)-receptor (PPAR-δ) complex is stable. This significant binding affinity and stable ligand-receptor complex suggests a high potential for pharmacological efficacy. While the compound's high TPSA and numerous hydrogen bond donors and acceptors may hinder its permeability and bioavailability, these characteristics also suggest a potential for strong and specific interactions with the target, which could enhance its therapeutic specificity and effectiveness.

Despite the challenges posed by its physicochemical properties, the strong binding affinity of Schaftoside (**B7**) underscores its potential as a therapeutic agent. Many successful drugs have been developed that violate Lipinski's rules but are effective due to their potent biological activity [45–47]. Therefore, Schaftoside (**B7**) merits further investigation and optimization. Strategies to improve its bioavailability and permeability, such as chemical modifications or advanced drug delivery systems, could unlock its potential. The promising binding affinity demonstrated by Schaftoside (**B7**) highlights the need for continued research to harness its therapeutic capabilities.

## 2.7. Analysis of passive membrane permeability and translocation pathways

The passive membrane permeability of pharmaceuticals can be determined experimentally using the PAMPA, which involves quantifying the substances that have moved across a flat artificial membrane from donor to acceptor wells [48]. Novel techniques have been created to track the entrance of pharmaceuticals into single-layered liposomes in real-time by utilizing fluorescent markers. The drug membrane permeability (P) is calculated as the ratio of the drug flow (J) through the membrane to the difference in drug concentrations between the donor (outside) and acceptor (inside) compartments, expressed as P = J/ΔC [49, 50].

The PerMM software was used to estimate the mean potential force profile (PMF) across the membrane, as shown in Figs 7 and 8, and determine the permeability coefficient across a dioleoyl-phosphatidylcholine (DOPC) bilayer membrane, as listed in Table 6, for the compounds under investigation (**B7** and **B12**). This method allows for the prediction of important physicochemical parameters associated with how chemicals interact with the lipid bilayer. The PMF analysis provides insights into the energy landscape of compound translocation across the membrane, while the permeability coefficient data delivers statistical data on their diffusion through the lipid bilayer [51]. Evaluating the compounds' potential application and efficiency in diverse biomedical and pharmaceutical contexts is made possible by these thorough examinations, which also greatly add to our understanding of the compounds' behaviour at the molecular level. The two phytoconstituents continued to the selection for membrane permeability on the basis of their significance binding affinity. The results showed that both the phytoconstituents (**B7** and **B12**) could penetrate the plasma membrane through passive diffusion. The energy transfer influences the membrane's permeability to a substance. Energy transfer is the amount of energy needed to move through the plasma membrane, and this value varies based on the compound's location in the lipid bilayer. Hydrophobic molecules exhibit the highest energy transfer values at the centre and the lowest at the ends of the lipid bilayer. Hydrophobic molecules have the lowest energy transfer at the center and the highest at the periphery [49].

The result in (Figs 28S to 31S, Tables 3S and 4S in S1 File) reveals that both phytochemicals had negative energy profiles consistently across all depths of the graph, with a stable state detected near the membrane's core. The chemicals analyzed have high ClogP values, indicating that they are likely to partition more into the hydrophobic inner area of the lipid bilayer rather than the lipid-water interface in the membrane. The energy minimum for both compounds is expected to be located at the point of contact with the initial lipid layer (about 2 nm). The

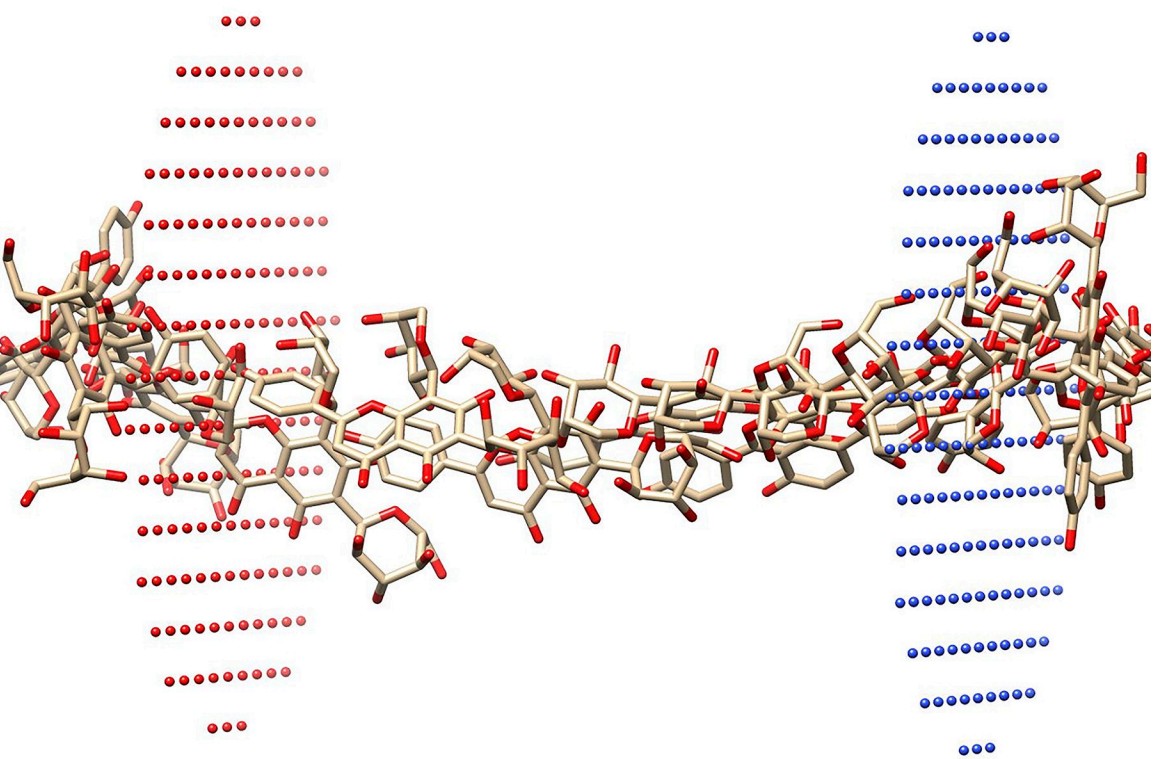

**Fig 7. Pathway of passive diffusion across the cell membrane for compound Schaftoside (B7).**

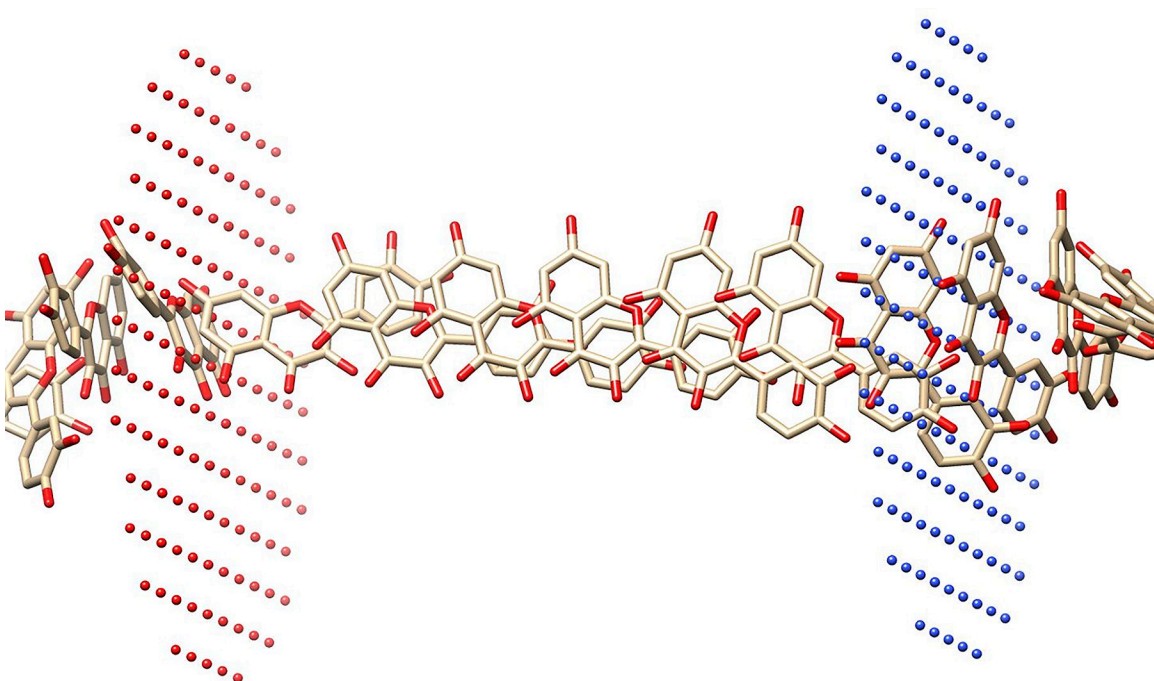

**Fig 8. Pathway of passive diffusion across the cell membrane for compound Quercetin (B12).**

**Table 6. Permeability predictions as calculated using the PerMM server.**

| SN | Phytoconstituents | Free energy of binding (DOPC) |
|---|---|---|
| **1.** | Schaftoside (**B7**) | -2.71 kcal/mol |
| **2.** | Quercetin (**B12**) | -2.00 kcal/mol |

binding free energy increases with the molecule's lipophilicity. After passing through the initial lipid layer, a new energy barrier must be overcome before reaching the subsequent lipid layer. Given that the energy profiles resemble those found in the compounds (menthol carbonates) described by Mollazadeh et al. [52], it is plausible that a "flip-flop" motion takes place in these compounds, enabling faster movements between layers and potentially achieving higher concentrations on the cytoplasmic side. PerMM's projected permeability coefficients confirm the preceding assumptions (Table 6).

## 2.8. HepG2 cell line study

The cytotoxic activity of *A. helferiana* extract (sample name Gai Khurey) was evaluated on HepG2 cell line by MTT assay. The $IC_{50}$ stands for half-maximal inhibitory concentration, which is a measure of the potency of a substance in inhibiting a biological or biochemical function by 50%. In this case, the $IC_{50}$ values are given in µg/mL, indicating the concentration at which each substance is able to inhibit the growth of HepG2 cells by 50% [53].This screening was conducted to validate the in-silico molecular docking results. *A. helferiana* extract exhibited an $IC_{50}$ value of 236.93 µg/mL, indicating that a relatively large concentration is required to inhibit 50% of the HepG2 cells. This suggests that the extract has moderate to low cytotoxicity against this liver cancer cell line, highlighting its potential as an anticancer agent, though it is considerably less potent compared to established treatments (Table 7 and Fig 9). In contrast, 5-Fluorouracil demonstrated a significantly lower $IC_{50}$ value of 5.0 µg/mL [54], indicating a strong cytotoxic effect and requiring only a small concentration to achieve 50% inhibition. Despite the higher $IC_{50}$ of *A. helferiana* extract, its moderate cytotoxicity suggests the presence of bioactive compounds with potential anticancer properties, warranting further investigation.

## 3. Conclusion

The current research provides the evidence of presence of 17 phytoconstituents such as Vicenin 1, Violanthin, Schafroside, Coumarin, Quercetin, Angiopteroside, Corosolic acid, etc in *A. helferiana* extract. It was reported the extract had a cytotoxic activity against HepG2 cell line by MTT assay. Therefore, we believe that these phytoconstituents may serve as a source for novel lead structures in the development of cancer-fighting agents. Furthermore, molecular docking, drug likeliness properties, and cell membrane permeability test revealed that the Quercetin (**B12**) had a high binding affinity for the target protein with best membrane permeability obeying the Lipinski's rule of five to be a drug candidate. However, the instability of

**Table 7. Anticancer activity of *Angiopteris helferiana* extract on HepG2 cell line by MTT assay.**

| SN | Sample | HepG2 cell line$IC_{50}$ in µg/mL |
|---|---|---|
| 1. | *Angiopteris helferiana* extract | 236.93 |
| 2. | 5- Fluorouracil | 5.0 [54] |

Note: "An extract is defined to be highly active if it has $IC_{50}$ < 10 µg/ml, active when the $IC_{50}$ is between 10 µg/ml and 100 µg/ml, moderately active if the $IC_{50}$ is between 100µg/ml and 500 µg/ml, and low activity if the $IC_{50}$ is > 500 µg/ml" [55].

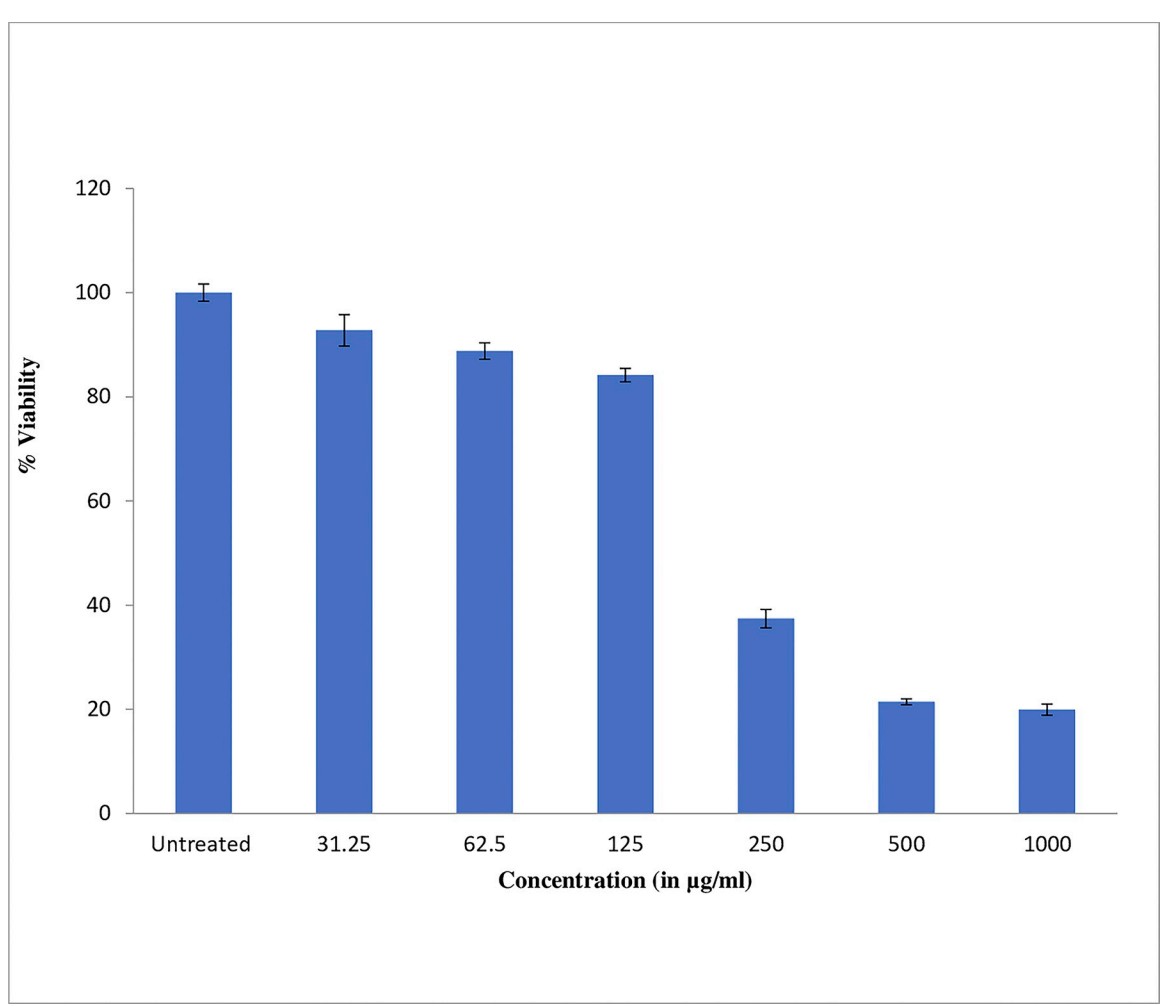

**Fig 9. HepG2 cell line vs *Angiopteris helferiana* extract.**

ligand-receptor complex hinders its potency as anti-cancer agents. Instead Schaftoside (**B7**) showed the best stability to binds with receptor and showed the significant potency. In conclusion, *A. helferiana* extract may be the effective therapeutic lead for anticancer drugs but it acquires further experimental validation.

## 4. Limitation of the study

We do not claim the precision and accuracy of interpretation of LC-MS data due to the unavailability of precise and accurate reference mass data. There is very limited research on LC-MS profiling for *A. helferiana*. Due to the lack of sophisticated lab, we were unable to isolate individual phytoconstituents from *A. helferiana* extract which may affected the accuracy of in-vitro result.

## 5. Materials and methods

### 5.1. Collection and authentication of plant

Fresh rhizomes of *A. helfiarana* were collected in May, 2023 from Vyas municipality, Tanahun district, Nepal (geographical coordinate = 27.989131, 84.287275, https://maps.app.goo.gl/xQ93UaMFqodA5ypZ8). Since the plant collection site was open to public, permission from

the municipality was not required. The authenticity of the plant specimen (31/02/2080/Herbarium specimen No) was confirmed at the National Herbarium and Plant Laboratories, Godawari-3, Lalitpur, Nepal, by referring the deposited plant specimen.

## 5.2. Extraction and isolation

The fresh *A. helfeianara* rhizome were cleaned using distilled water, reduced its size using plant cutter, and shade dried for one week to remove moisture content. Successive extraction was done using the methodology of previous research with slight modifications [16]. *A. helfeirana* rhizome (100 g) was subjected to successive extraction using 70% methanol as solvent in the ratio of 1:10 w/v at 55˚C for 2 hours followed by room temperature for 22 hours, and 70% methanol at room temperature for 24 hours. Extracts were filtered using a thick cotton bed and combined to evaporate under reduced pressure at 55˚C using rotary evaporator (Biobase RE-2000B, German). The concentrated extracts so obtained after evaporation were poured in beaker, followed by complete drying in vacuum desiccators. All the dried extracts were weighted and stored at 4˚C until further use. Extractive yield was estimated as:

% Extractive yield = (weight of dry extract / initial weight of dry sample) × 100

The yield obtained from *A. helferiana* was 22.81% w/w.

## 5.3. FT-IR screening

FT-IR provides the insight of functional group present in sample. The department of central instrumental analysis, Nargund College of Pharmacy, Bengaluru, India provided the space and facility for spectral analysis. The IR spectra of methanolic extract was recorded on a FT-IR (Model Shimadzu 8700) in the range of 400–4000 cm$^{-1}$ by KBr pellet method [56].

## 5.4. LC-MS profiling

In the process of conducting Liquid Chromatography-Mass Spectrometry (LC-MS) analysis of a plant extract in a sophisticated instrumental laboratory, a series of sequential steps were followed. We performed LC/MS in the sophisticated analytical instrumentation facility, Honeychem pharma research pvt ltd, Bangalore, India. The liquid chromatographic system was UPLC Acquity H class series system where separations were achieved on a WATERS XBridge (50 X 4.6mm 3.5μ), $C_{18}$ column. The mobile phase consisted of 0.1% formic acid in water (A) and 0.1% formic acid in Acetonitrile HPLC grade (B) at a flow rate of 1.2 mL/min. The analysis was performed using the gradient elution & the condition applied were represented in Tables 1S and 2S in S1 File.

Each extract sample was analysed in positive mode. Identification of secondary metabolites in the extracts of *A. helferiana* was exclusively based on LC retention time and high- resolution mass spectra. Moreover, the estimation of the elemental composition of ions was based on registered MS m/z values. MS modes were performed with Triple Quadrupole (QqQ) MS/MS analysers in the positive mode [57]. Data, including mass spectra and retention times, were acquired from the mass spectrometer and subsequently analyzed using MassLynxV4.1SCN805 software. Compound identification was performed by comparing mass spectra and retention times with reference standards, published data, and various databases web server.

## 5.5. Molecular docking protocol

**5.5.1. Selection of ligand and protein.**   In this molecular docking study, the protein target selected was human peroxisome proliferator-activated receptors (PPAR-δ), specifically in complex with the agonist AZ 242 (PDBID: 1I7G) [58]. PPAR-δ is crucial in the

**Fig 10. 2D structure of phytoconstituents along with their name and code (B1 toB17).**

pathophysiology of HCC along with the role in metabolic process [59]. This is the novel and most promising receptor for the development of anti-cancer agents. Various clinical observation reported that, the epidermal growth factor receptor (EGRF) is suspectable to develop resistance against drugs [60]. But, PPARs are ligand-activated transcription factors that are part of the nuclear hormone receptor superfamily, playing crucial roles in regulating energy balance, cell differentiation, proliferation, apoptosis, and inflammation. Initial studies connected PPARs to the development of cancer, and currently, PPARs are associated with various solid organ cancers, including those of the breast, ovary, prostate, bladder, stomach, colon, and lungs. Therefore, we choose PPAR- δ for our research as receptor [61, 62]. The 3D structure of the protein was obtained from the Protein Data Bank (PDB) (https://www.rcsb.org/) and subjected to purification using BIOVIA Discovery Studio 2021 to eliminate water molecules, heteroatoms, and undesired ligands. Subsequently, the protein structure underwent optimization, including energy minimization, to prepare it for the docking protocol. The protein was selected on the basis of biological relevance, availability of a known 3D structure (resolution = 2.20 Å), and relevance to the research question (non-mutated). The chosen protein was further optimized for docking studies, ensuring its suitability for the investigation. Simultaneously, 17 phytoconstituents derived from LC-MS analysis of *A. helferiana* were chosen as ligands (Fig 10). Their 3D structures were retrieved from the Indian Medicinal Plants, Phytochemistry and Therapeutics 2.0 (IMPPAT 2.0) database (https://cb.imsc.res.in/imppat/) [63] in simple dimension file (SDF) format, followed by energy minimization (MMFF 96 force field), chirality optimization, and conversion into PDBQT format from Open babel 2.4.1 software [64].

**5.5.2. Validation of protein.** The PROCHECK tool (http://saves.mbi.ucla.edu/) [65] was used to determine the accuracy and quality of the selected PPAR-δ protein. Moreover, the physiochemical properties of selected protein were determined by Expasy-ProtParam web server (https://web.expasy.org/protparam/) [66].

**5.5.3. Grid box generation and docking process.** The molecular docking simulation was conducted using AutoDock Vina1.5.7 software [67], wherein the prepared PPAR-δ protein

and 17 phytoconstituents (ligands) structures were loaded, and docking parameters were set, encompassing grid box dimensions, search parameters, and scoring functions. The grid box generation in AutoDock Vina 1.5.7 involves setting up the parameters that define the search space within which the docking simulations will take place. In this case, the default settings for AutoDock Vina 1.5.7 were utilized, where the grid box axis and dimensions were automatically selected. The default parameters for the grid box were as follows: x-axis = 30.857 Å, y-axis = 29.068 Å, and z-axis = 30.740 Å, and the overall size of the grid box was set to 40 Å. The specified dimensions (30.857 Å, 29.068 Å, and 30.740 Å) define the lengths of the grid box along each axis, while the overall size of 40 Å suggests the cubic shape of the grid box. This grid box serves as the exploration space for the ligand to dock into the binding site of the protein [68]. The dimensions and size of the grid box are crucial parameters, influencing the accuracy and efficiency of the docking simulation. AutoDock's default settings, as described, were applied in this procedure to ensure a balanced and appropriate exploration of the protein's active site by the ligands during the docking studies. The software explored potential binding conformations, predicting the binding affinity of each ligand with the protein. Post-simulation, the results were meticulously analyzed, focusing on binding affinity and the mode of interaction between ligands and the protein. The evaluation was carried out with emphasis on docking scores and interaction patterns to identify the most promising ligands [69].

**5.5.4. Results visualization.** The visualization of the docked complex was accomplished using BIOVIA Discovery Studio Visualizer 2021 software, facilitating the analysis of binding modes, hydrogen bonding, and other crucial interactions between phytoconstituents (ligands) and the PPAR-δ protein. The results obtained from this study were used in further experiments to validate and explore the biological significance of the predicted interactions [70].

**5.5.5. Docking results validation.** To validate docking results using PyMol 2.5.2 software, the procedure involved loading both the co-crystal native ligand and docked complex structures, superimposing the structures for alignment, and calculating the root mean square deviation (RMSD) between the docked ligand and the reference co-crystal ligand. The PyMol 2.5.2 command for RMSD calculation was used to obtain quantitative information on structural similarity. The results were visualized in PyMol 2.5.2, where a lower RMSD indicates better alignment and more accurate prediction. Typically, an RMSD below 2 Å is considered good, while values above 4 Å may suggest less reliable predictions (Fig 1S in S1 File). The entire analysis, including specific binding interactions and spatial orientation, was checked for comprehensive validation of the molecular docking results [71].

## 5.6. ADME prediction

To determine absorption, distribution, metabolism, and excretion (ADME) properties, we employed the ADMETlab 2.0 server (https://admetmesh.scbdd.com/) [72], a computational platform for predicting drug-like properties. The procedure involved accessing the ADMETlab 2.0 server through a web browser. Once on the platform, we input the simplified molecular-input line-entry system (SMILES) notation of the compound of interest. The server employed various algorithms and multi-task graph attention framework along with batch computation module to predict key ADME properties. The absorption prediction assesses factors like gastrointestinal absorption, partition coefficient and water solubility. Distribution predictions consider factors influencing the compound's distribution within the body. Metabolism predictions evaluate the likelihood of metabolism by cytochrome P450 enzymes. Excretion predictions focus on renal and non-renal clearance [73].

**5.6.1. Passive membrane permeability and translocation pathways.** For the purpose of performing quantitative analysis and visualizing the passive translocation of bioactive

compounds across lipid membranes, the PerMM web server (https://permm.phar.umich.edu/) [74] was set up. The server is a physics-based web application that calculates membrane binding energies and permeability coefficients of various compounds across different types of membranes such as phospholipid bilayers, parallel artificial membrane permeability assay-double sink (PAMPA-DS), blood-brain barrier (BBB), and $CaCO_2$/MDCK (Madin–Darby canine kidney cell line) cell membranes. The process visualizes the movement of a substance across a lipid bilayer by showing its translational and rotational locations, as well as the changes in solvation energy [75].

### 5.7. In-vitro cell line study

Each well of a 96-well microtiter plate was filled with 200 μL of a medium (HepG2 Human liver cancer cell line) containing roughly 10 thousand cells. Additionally, 200 μL of five different concentrations of test materials were added to each well. 5-Fluorouracil was used as the standard. After adding 10% MTT (3-(4, 5-dimethylthiazolyl-2)-2, 5-diphenyltetrazolium bromide) reagent to each well to reach a concentration of 0.5 mg/mL, the samples were incubated for 3 hours. The culture media was removed without disturbing the formed crystal. The formazan was subsequently dissolved by adding 100 μL of DMSO (a solubilization solution) and gently agitating it in a gyrator shaker. The time of contact used between extract sample with the cell line was 24 hours. The measurement of absorbance was conducted at wavelengths of 570 nm and 630 nm [76].

## Supporting information

**S1 File.**
(DOCX)

## Acknowledgments

We thank Honeychem pharma research Pvt. Ltd., Bangalore, India, Nargund College of Pharmacy and Pokhara University for providing the site of instrumental analysis. We also thank CellKraft Biotech Pvt. Ltd, Banglore, India for providing cytotoxic activity.

## Author Contributions

**Conceptualization:** Bipindra Pandey, Shankar Thapa.

**Data curation:** Bipindra Pandey, Shankar Thapa, Jaya Bahadur Ghale, Ram Kishor Yadav, Poojashree V.

**Formal analysis:** Bipindra Pandey, Shankar Thapa, Mahalakshmi Suresha Biradar, Bhoopendra Singh, Jaya Bahadur Ghale, Pramod Kharel, Sujan Dawadi.

**Investigation:** Pramod Kharel, Prabhat Kumar Jha.

**Methodology:** Bipindra Pandey, Shankar Thapa, Mahalakshmi Suresha Biradar.

**Resources:** Bipindra Pandey, Jaya Bahadur Ghale, Prabhat Kumar Jha, Ram Kishor Yadav.

**Software:** Shankar Thapa, Mahalakshmi Suresha Biradar.

**Supervision:** Shankar Thapa.

**Validation:** Bipindra Pandey, Mahalakshmi Suresha Biradar.

**Visualization:** Bipindra Pandey, Mahalakshmi Suresha Biradar, Jaya Bahadur Ghale, Pramod Kharel.

**Writing – original draft:** Bipindra Pandey, Shankar Thapa.

**Writing – review & editing:** Bhoopendra Singh, Jaya Bahadur Ghale, Pramod Kharel, Prabhat Kumar Jha, Ram Kishor Yadav, Sujan Dawadi, Poojashree V.

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
