## [Decision Letter · Decision Letter 0]

26 Jun 2024

PONE-D-24-20408LC-MS Profiling and Cytotoxic Activity of Angiopteris helferiana Against HepG2 Cell Line: Molecular Insight to Investigate Anticancer AgentPLOS ONE

Dear Dr. Thapa,

Thank you for submitting your manuscript to PLOS ONE. After careful consideration, we feel that it has merit but does not fully meet PLOS ONE’s publication criteria as it currently stands. Therefore, we invite you to submit a revised version of the manuscript that addresses the points raised during the review process.

We look forward to receiving your revised manuscript.

Kind regards,

Ghulam Mustafa, PhD

Academic Editor

PLOS ONE

Journal Requirements:

Reviewers' comments:

Reviewer's Responses to Questions

**Comments to the Author**

1. Is the manuscript technically sound, and do the data support the conclusions?

Reviewer #1: Partly

Reviewer #2: Yes

Reviewer #3: Partly

2. Has the statistical analysis been performed appropriately and rigorously? 

Reviewer #1: N/A

Reviewer #2: I Don't Know

Reviewer #3: N/A

3. Have the authors made all data underlying the findings in their manuscript fully available?

Reviewer #1: Yes

Reviewer #2: Yes

Reviewer #3: Yes

4. Is the manuscript presented in an intelligible fashion and written in standard English?

Reviewer #1: Yes

Reviewer #2: Yes

Reviewer #3: Yes

5. Review Comments to the Author

**Reviewer #1:** The work is oriented towards the anticancer effect of exttract of Angiopteris helferiana (A. helferiana) C. Presl is a large 88 fleshy fern. However it is traditionaly used in many diseases, I have to reaise a question on what base the authors selected the anticancer against one line of cells? is it indicative results to get the results against HepG2 Cell only?Also, the activity due to the extract however the authors recorded the activity is due to very well known compounds like quercetion where there are a lot of reports regardong this compound as anticancer and up till now not used, So, extensive study on these componds is consider a waste of time and effeort.

I think the authors have to work in purified fraction preceded with activity guided fractionation to get standartized active simple fraction could be applied in pharmaceutical market.

-table 1- column 10 it is coumarin not falavonoid because coumarin is 9 carbon mostly and flavonoid is 15 carbon

Please revise the content vey well.

discussion need more information to get the conclusion in right way.

**Reviewer #2: **1. the authors must compare the effect of cyto-toxic on normal cell line like RPE1 or BJ1

2. the IC 50 of 5-fluorouracil is too high

3. the time of contact between extract or drug with the cell is not mentioned

**Reviewer #3:** . PLOS ONE

LC-MS Profiling and Cytotoxic Activity of Angiopteris helferiana Against HepG2 Cell Line: Molecular Insight to Investigate Anticancer Agent

.

Reviewer comments:

Results and discussion: lines111/112: peak at 2927cm-1 confirms the presence of phenolic group? The band 2927cm-1 is attributed the aliphatic C-H stretching, revise the value again.

Lines 116/117: 1392 cm-1and 1269 cm-1indicates the involvement of stretching vibration of different aliphatic alkyl groups.?? The given 1392 and 1269cm-1 values are not due to C-H stretching? Please change the assignment values again.

2.4 Molecular Docking: line: 174

A. helferiana extract) when bound to a target molecule (human PPAR-δ receptor).

Why did the author choose Humn PPAR-δ as the enzyme-protein target for A.helferiana more than alternative binding targets? Like EGFR and VEGFR? HDACs, etc. Provide reasonable explanations.

Furthermore, did the author do docking binding studies with other targets than PPAR-δ, and verified that PPAR-δ is the best for interaction according to binding free energy (Kcal/mol) and RMSD (Ẩ)?

The author provided a thorough description of the interactions between the secondary metabolites and the target protein, including the sorts of interactions.

2.7. Analysis of passive membrane permeability and translocation pathways, line:328: compounds (Menthol carbonates) described by Mollazadeh et al, cite reference number for this author.

2.8.HepG2 cell line study

Lines: 338-340 :This screening was conducted to validate the in-silico molecular docking results. Interestingly, the extract showed the significant cytotoxic activity having IC50 value of 236.93 μg/ml (Table 7 and Figure 9). This result indicates the potential activity of extract towards HepG2 Human liver cancer cell line.>>> The IC50 value of 236.93µg/ML for the extract is low-moderate and not notable

NOTE: An extract is defined to be highly active if it has IC50 < 10 µg/ml, active when the IC50 is between 10 µg/ml and 100 µg/ml, moderately active if the IC50 is between 100µg/ml and 500µg/ml, and low activity if the IC50 is ˃ 500 µg/ml

Bahadori, M H., Azari, Z., Zaminy, A., Dabirian, S., Mehrdad, S M., & Kondori, B J. (2021, March 31). Anti-proliferative and apoptotic effects of hull-less pumpkin extract on human papillary thyroid carcinoma cell line. , 54(1), 104-111. https://doi.org/10.5115/acb.20.22.

The author should test the cytotoxicity of the extract on a normal cell line, and then calculate the selectivity index (SI) (IC50 normal cells / IC50 cancer cells).

Line: 342: The reference (35): The drug 5-Flurouracil, IC50= (435.59 μg/ml,(35)), and Table 7:>>> This reference specifies the IC50 against the HCT116 cell line,?? NOT against the HEPG2 cancer cell line; correct the statement and conclusion, which do not match the HEPG2 cell line that you employed.

Reference: must be written as references.

** It was obvious from the 17 phytoconstituents that each secondary metabolite was docked with the target enzyme, although the IC50 was measured collectively for the entire extract, due to the difficulty in isolating each constituent alone.

In vitro evaluation for secondary metabolites requires particular tests, such as cell cycle arrest, apoptosis, and enzyme assays. However, the difficulty of separating individual phytoconstituents limits the study.

……………………………………………END………………………………………………………

6. PLOS authors have the option to publish the peer review history of their article (what does this mean?). If published, this will include your full peer review and any attached files.

Reviewer #1: **Yes: **Prof. Abdel Nasser B. Singab, Ain Shams University, Faculty of Pharmacy, Dept. of Pharmacognosy, Founder and supervispor of Center of Drug Discovery Research and Develpment

Reviewer #2: No

Reviewer #3: **Yes: **Ammar A. Razzak Mahmood

---

## [Author Response · Author response to Decision Letter 0]

1 Jul 2024

We would like to thank reviewer for their valuable insight and comments. we have provided the word file stating point-wise response the reviewers.

we have revised the manuscript as stated by editor and added the information asked by journal.

---

## [Decision Letter · Decision Letter 1]

20 Aug 2024

LC-MS Profiling and Cytotoxic Activity of Angiopteris helferiana Against HepG2 Cell Line: Molecular Insight to Investigate Anticancer Agent

PONE-D-24-20408R1

Dear Dr. Thapa,

We’re pleased to inform you that your manuscript has been judged scientifically suitable for publication and will be formally accepted for publication once it meets all outstanding technical requirements.

Kind regards,

Ghulam Mustafa, PhD

Academic Editor

PLOS ONE

Additional Editor Comments (optional):

Reviewers' comments:

Reviewer's Responses to Questions

**Comments to the Author**

1. If the authors have adequately addressed your comments raised in a previous round of review and you feel that this manuscript is now acceptable for publication, you may indicate that here to bypass the “Comments to the Author” section, enter your conflict of interest statement in the “Confidential to Editor” section, and submit your "Accept" recommendation.

Reviewer #3: All comments have been addressed

2. Is the manuscript technically sound, and do the data support the conclusions?

Reviewer #3: Yes

3. Has the statistical analysis been performed appropriately and rigorously? 

Reviewer #3: Yes

4. Have the authors made all data underlying the findings in their manuscript fully available?

Reviewer #3: Yes

5. Is the manuscript presented in an intelligible fashion and written in standard English?

Reviewer #3: Yes

6. Review Comments to the Author

Reviewer #3: The author addressed the comments made in the prior contribution.

The compound's docking profile is explored and fully described.

In addition, the IC50 value of 5-FU was adjusted, and the author conducted a thorough comparison with the recovered bioactive compounds.

7. PLOS authors have the option to publish the peer review history of their article (what does this mean?). If published, this will include your full peer review and any attached files.

Reviewer #3: **Yes: **Ammar A. Razzak Mahmood

---

## [Editor Report · Acceptance letter]

9 Sep 2024

PONE-D-24-20408R1 

PLOS ONE

Dear Dr. Thapa, 

I'm pleased to inform you that your manuscript has been deemed suitable for publication in PLOS ONE. Congratulations! Your manuscript is now being handed over to our production team.

Kind regards, 

on behalf of

Dr. Ghulam Mustafa 

Academic Editor

PLOS ONE